# Evaluating Phytochemical Profiles, Cytotoxicity, Antiviral Activity, Antioxidant Potential, and Enzyme Inhibition of *Vepris boiviniana* Extracts

**DOI:** 10.3390/molecules28227531

**Published:** 2023-11-10

**Authors:** Kassim Bakar, Andilyat Mohamed, Łukasz Świątek, Benita Hryć, Elwira Sieniawska, Barbara Rajtar, Claudio Ferrante, Luigi Menghini, Gokhan Zengin, Małgorzata Polz-Dacewicz

**Affiliations:** 1Laboratoire Aliments, Réactivité et Synthèse des Substances Naturelles, Faculté des Sciences et Techniques, Université des Comores, Moroni 167, Comoros; karibu5051@gmail.com; 2Department of Pharmacy, Botanic Garden “Giardino dei Semplici”, Università degli Studi “Gabriele d’Annunzio”, Via Dei Vestini 31, 66100 Chieti, Italy; nelofarkhattak@gmail.com (N.); claudio.ferrante@unich.it (C.F.); luigi.menghini@unich.it (L.M.); 3Physiology and Biochemistry Laboratory, Department of Biology, Science Faculty, Selcuk University, Konya 42130, Turkey; 4Herbier National des Comores, Faculté des Sciences et Techniques, Université des Comores, Moroni 167, Comoros; andilyat.mohamed@gmail.com; 5Department of Virology with Viral Diagnostic Laboratory, Medical University of Lublin, Chodźki 1, 20-850 Lublin, Poland; barbara.rajtar@umlub.pl (B.R.); malgorzata.polz-dacewicz@umlub.pl (M.P.-D.); 6Department of Natural Products Chemistry, Medical University of Lublin, Chodźki 1, 20-093 Lublin, Poland; benytka@gmail.com (B.H.); esieniawska@pharmacognosy.org (E.S.)

**Keywords:** *Vepris*, antioxidant, LC-MS, antiviral, cytotoxic, natural products

## Abstract

In the present study, we performed comprehensive LC-MS chemical profiling and biological tests of *Vepris boiviniana* leaves and stem bark extracts of different polarities. In total, 60 bioactive compounds were tentatively identified in all extracts. The 80% ethanolic stem bark extract exhibited the highest activity in the ABTS assay, equal to 551.82 mg TE/g. The infusion extract of stem bark consistently demonstrated elevated antioxidant activity in all assays, with values ranging from 137.39 mg TE/g to 218.46 mg TE/g. Regarding the enzyme inhibitory assay, aqueous extracts from both bark and leaves exhibited substantial inhibition of AChE, with EC_50_ values of 2.41 mg GALAE/g and 2.25 mg GALAE/g, respectively. The 80% ethanolic leaf extract exhibited the lowest cytotoxicity in VERO cells (CC_50_: 613.27 µg/mL) and demonstrated selective cytotoxicity against cancer cells, particularly against H1HeLa cells, indicating potential therapeutic specificity. The 80% ethanolic bark extract exhibited elevated toxicity in VERO cells but had reduced anticancer selectivity. The n-hexane extracts, notably the leaves’ n-hexane extract, displayed the highest toxicity towards non-cancerous cells with selectivity towards H1HeLa and RKO cells. In viral load assessment, all extracts reduced HHV-1 load by 0.14–0.54 log and HRV-14 viral load by 0.13–0.72 log, indicating limited antiviral activity. In conclusion, our research underscores the diverse bioactive properties of *Vepris boiviniana* extracts, exhibiting potent antioxidant, enzyme inhibitory, and cytotoxicity potential against cancer cells.

## 1. Introduction

Research in the field of bioactive plant molecules has become a priority in many countries around the world. Researchers and consumers’ interest in natural products, particularly plant-based products, has become a global trend in the face of new emerging diseases. Indeed, in recent decades, numerous research studies have highlighted the unexpected side effects of synthetic drugs. This has further strengthened the prospects for research into plants of pharmaceutical interest worldwide [1,2,3]. Herbal medicines are often preferred because of their predictable, minimal, or lack of side effects and their combinations of biologically active components with minerals and vitamins, which have advantages over synthetic medicines [2]. The constantly increasing demand for herbal products has encouraged phytochemical and pharmacological research on plants, mainly based on ethnopharmacological knowledge [4].

The genus *Vepris* Comm. ex A. Juss. (Rutaceae) comprises approximately 100 species according to different botanical databases, namely TROPICOS, the International Plant Names Index, and the Plant List [5,6]. It consists of shrubs and trees, widespread mainly in tropical Africa, Zanzibar, Madagascar, Comoros, and the Mascarene Islands (Indian Ocean Islands) [7]. The Indian Ocean is home to many biodiversity hotspots, as 25% of the world’s biodiversity can be found there, as well as in Sub-Saharan Africa [8]. For this genus, 25 species have been recorded in Madagascar alone, including 24 plants endemic to the island [9]. *V. boiviniana* is native to the Comoros and Madagascar. It is found at different altitudes on both sides of the two countries. Generally, the plant has different uses in different parts of the Indian Ocean and its surroundings. In addition to its use as a euphoric plant, a leaf infusion is taken as an astringent. However, the Comoros, *V. boiviniana* is mainly used against redness of the skin. In our ethnobotanical field work, we learned that leaves are also used to make herbal tea [5,8,10]. 

In this study, our objective was a comprehensive screening of various extracts (including n-hexane, ethanol/water (80%), and aqueous extracts (macerated and infused)) derived from the leaves and stem bark of *V. boiviniana*. We assessed their antioxidant, anti-enzymatic, cytotoxic, and antiviral activities and performed LC-MS chemical characterization. To evaluate antioxidant effects, we employed a range of chemical methods, including ABTS, DPPH, CUPRAC, FRAP, phosphomolybdenum, and MCA. For enzyme inhibitory properties, we examined their effects on cholinesterases, amylase, glucosidase, and tyrosinase. Cytotoxicity was assessed using non-cancerous and various cancerous cells, and we evaluated antiviral potential against HHV-1 (Human Herpesvirus type 1) and HRV-14 (Human Rhinovirus type 14). The results obtained from this study provide a scientific starting point for further research on *V. boiviniana*.

## 2. Results and Discussion

### 2.1. Chemical Characterization

The phytochemical analysis (Table 1) of the studied extracts revealed the presence of compounds belonging to several chemical classes. Organic acids were represented by arabinoic, gluconic, quinic, malic, citric, and azaleic acids, while phenylethanoids were represented by hydroxytyrosol, saliroside (glucoside of tyrosol), and hydroxythyrosol glucoside. In the class of phenolic acids, dihydroxybenzoic acid, caffeic acid, coumaroylquinic acid, isoferulic acid, hydroxybenzoic acid isomers, and isomers of caffeoylquinic acid and *di*-caffeoylquinic acid were identified. Two alkaloids were described, the masses of which can be tentatively assigned to tecleabine/norisoboldine (313 Da) and tecleabine derivative/boldine (327 Da). However, because of the lack of authentic standards and high-resolution mass spectra available in the literature sources, we were not able to precisely assign these compounds. Flavonoids were present in quite a large number, as aglycones (epicatechin, taxifolin, apigenin, isorhamnetin) and glycosides (quercetin 3-*O*-rutinoside-7-*O*-glucoside, kaempferol 3-*O*-rutinoside-7-*O*-glucoside, isorhamnetin 3-*O*-rutinoside-glucoside, quercetin 3-*O*-rutinoside-pentoside, quercetin *O*-glucoside *O*-pentoside, rutin, quercetin 3-*O*-glucoside, kaempferol 3-*O*-rutinoside, isorhamnetin 3-*O*-rutinoside, quercetin *O*-pentoside, quercetin 3-*O*-glucuronide, quercetin 3-*O*-rhamnoside, isorhamnetin 3-*O*-glucoside, quercetin 3-*O*-caffeoyl-glucoside, and kaempferol 3-*O*-rhamnoside). A few tannins were described, including procyanidin dimers and trimers. Several carbohydrates and skeletons of norisoprenoid glucosides and terpene glucosides were also detected.

Carbohydrates and tannins were more abundantly detected in bark extracts, while norisoprenoid glucosides and terpene glucosides were found in leaves. Additionally, glycosides of quercetin, kaempferol, and isorhamnetin were predominately found in leaf extracts. The following compounds were detected only in leaves: quercetin 3-*O*-rutinoside-7-*O*-glucoside, kaempferol 3-*O*-rutinoside-7-*O*-glucoside, isorhamnetin 3-*O*-rutinoside-glucoside, quercetin 3-*O*-rutinoside-pentoside, quercetin *O*-glucoside *O*-pentoside, kaempferol 3-*O*-rutinoside, isorhamnetin 3-*O*-rutinoside, and isorhamnetin 3-*O*-glucoside.

### 2.2. Total Phenolic–Flavonoid Content

The successful extraction of bioactive compounds from plant matrices relies heavily on the choice of the extraction solvent and the extraction method employed. The selection of the appropriate solvent and extraction technique plays a critical role in determining the efficiency and yield of bioactive compound extraction. These choices are pivotal in ensuring the effective isolation of valuable compounds from the plant material [22]. The results for the total phenolic and total flavonoid content are presented in Table 2. Upon comparison of the leaf and stem extracts, it is evident that the stem extracts exhibited notably higher phenolic contents, ranging between 27 and 108 mg GAE/g. Conversely, the leaf extracts displayed a higher concentration of flavonoids, with levels ranging from 13 to 25.73 mg RE/g. Among the various extracts, the ethanolic (80%) extract from stem bark demonstrated the highest phenolic content 108.19 mg GAE/g. Following closely behind, the infusion extract recorded a content of 75.85 mg GAE/g, while the ethanolic (80%) extract from leaves contained 70.61 mg GAE/g. In contrast, the infusion extract from leaves displayed a slightly lower phenolic content at 46.44 mg GAE/g. The aqueous extracts from stem bark and leaves yielded contents of 42.27 mg GAE/g and 34.98 mg GAE/g, respectively. The n-hexane leaf extract exhibited a relatively lower phenolic content compared to other extraction solvents, measuring 29.43 mg GAE/g. Nevertheless, it is noteworthy that this value still exceeded the phenolic content found in the n-hexane extract from stem bark, which was determined to be 27.49 mg GAE/g. 

These findings align with the LC-MS data, which indicate that bark contained higher levels of phenolics compared to the leaf extracts. Similar observations were reported in studies assessing the efficacy of hydroalcoholic solutions in extracting phenolics from various plant sources [22,23,24]. For instance, a substantial quantity of phenolics, measuring 626.34 mg GAE/100DW, was successfully extracted from *Vepris heterophylla*. However, it is worth noting that phenolics and flavonoids were undetectable in the hexane extract [25]. In contrast, the ethanolic root bark extract of *V. nobilis* primarily consisted of fatty acids (45.08%), sesquiterpenes (40.38%), and furaquinoline alkaloids [26]. These results corroborate earlier research that also found n-hexane extracts to have lower phenolic content [27]. It is evident that extracts obtained using n-hexane exhibited relatively lower total phenolic concentrations when compared to the extracts obtained using other solvents. This disparity can be attributed to the inherent properties of these solvents, which possess a diminished affinity for phenolic compounds [28,29]. According to the research conducted by Ntchapda et al., the hexane extract method demonstrated effectiveness in recovering essential oil from the *Vepris heterophylla* species [30]. This observation underscores the utility of hexane extraction in isolating valuable essential oils from this particular plant species.

Flavonoids are a class of polyphenolic plant compounds abundantly present in a wide array of fruits, vegetables, and select beverages. These substances, which have become known for their remarkable antioxidant abilities, are essential for cellular defense by actively scavenging free radicals and reducing the possibility of oxidative damage [31]. The importance of flavonoids in our diets and their potential advantages for human health are highlighted by the fact that this natural defense mechanism is essential for maintaining the structural integrity and general health of cells.

When considering the flavonoid content in leaves, it is notable that both the 80% ethanolic and infusion extracts exhibited higher and comparable quantities, each containing 25 mg RE/g. Subsequently, the n-hexane extract contained 20.73 mg RE/g, while the water extract recorded 13.12 mg RE/g. This same order of concentration was observed in the stem bark extract, with the 80% ethanolic extract displaying the highest flavonoid content measured at 6.13 mg RE/g, followed by the infusion extract containing 4.11 mg RE/g, the n-hexane extract at 3.42 mg RE/g, and the water extract at 1.17 mg RE/g. These findings are in concurrence with the LC-MS data, as depicted in Table 1, which unequivocally indicates that leaves possess a higher concentration of flavonoids compared to bark extracts. Furthermore, Table 1 underscores that the n-hexane extract, obtained from both stems and leaves, is notably deficient in several critical flavonoid compounds. Prior research conducted by Kiplimo et al. [32] provides additional support for these observations. Their study on *Vepris glomerata* demonstrated an even higher abundance of flavonoids, including the identification of novel flavonoid compounds. Within the extracts of the genus *Vepris*, flavonoids are consistently recognized as the third most prevalent component, trailing behind alkaloids and terpenoids [31,33]. These findings collectively emphasize the significance of flavonoids in the composition of *Vepris* extracts and contribute to our understanding of the distinctive chemical profile within this genus.

### 2.3. Antioxidant Properties

The most potent extracts, as determined by both free radical and reducing power assays, consistently contained the highest levels of total phenolics in both the leaf and bark extracts, specifically those prepared using 80% ethanol. In a range of antioxidant assays, with the exception of the metal chelating assay, the 80% ethanolic stem bark extract consistently exhibited superior antioxidant activity, with values ranging from 320 mg TE/g to 551 mg TE/g in free radical and reducing power assays. Remarkably, the 80% ethanolic stem bark extract displayed the highest activity in the ABTS assay, recording a value of 551.82 mg TE/g. It was closely followed by the CUPRAC assay, which yielded a value of 512.03 mg TE/g. Additionally, the DPPH assay demonstrated an antioxidant activity of 338 mg TE/g for this extract, while the FRAP assay demonstrated a value of 320.28 mg TE/g. Furthermore, the PBD assay exhibited activity of 2.41 mmol TE/g. These results collectively underscore the exceptional antioxidant potential of the 80% ethanolic stem bark extract, highlighting its capacity to effectively combat free radicals and oxidative stress in various assay systems. Following the 80% ethanolic extract, the infusion extract of stem bark consistently exhibited higher antioxidant activity across all tested assays, with values ranging from 137.39 mg TE/g to 218.46 mg TE/g. Notably, the infusion stem bark extract displayed the highest metal chelating activity among all extracts, with a value of 29.54 mg EDTAE/g, as depicted in Table 3. In the PBD assay, the most substantial value was obtained from the n-hexane leaf extract, measuring 2.31 mmol TE/g, closely followed by the 80% ethanolic extract of bark at 2.41 mmol TE/g. These results illuminate the varied antioxidant potential of different extracts, with the infusion extract standing out as particularly effective in scavenging free radicals and exhibiting strong metal chelating properties, highlighting its promising role in antioxidant applications. Following the 80% ethanolic and infusion bark extracts, the 80% ethanolic extract of leaves demonstrated the third-highest antioxidant activity overall. However, it is noteworthy that among the leaf extracts prepared with various solvents, the 80% ethanolic leaf extract displayed the most potent antioxidant activity. Interestingly, n-hexane extracts, both from leaves and bark, consistently displayed the lowest antioxidant activity in both free-radical and reducing-power assays. These results collectively indicate that the n-hexane extract possesses a moderate level of antioxidant activity. Our findings are consistent with those of Acquaviva and colleagues, who also found that the antioxidant activity of the n-hexane extract was lower than that of other extracts [34]. This implies that the n-hexane extract could be considered for utilization in situations where a moderate degree of antioxidant efficacy is sought, addressing the specific requirements of industrial or research applications. The wide spectrum of antioxidant activity observed among the different extracts highlights the significance of carefully choosing the extraction solvents that align with the intended application and the desired degree of antioxidant effectiveness. 

### 2.4. Enzymes Inhibition Properties

Enzyme inhibition is linked to the management/prevention of global health problems such as type II diabetes, Alzheimer’s disease, and obesity [35]. For example, acetylcholinesterase (AChE) catalyzes the breakdown of acetylcholine (ACh) in the synaptic cleft. In Alzheimer’s patients, ACh levels are lower than in healthy people. From this point on, inhibiting AChE can increase ACh levels in Alzheimer’s patients and thus increase cognitive functions [36]. Likewise, α-amylase and α-glucosidase are the most important enzymes for the hydrolysis of carbohydrates. In this context, their inhibition can regulate blood sugar levels in diabetics following a high-carbohydrate diet [37]. As another example, tyrosinase is a key enzyme in the synthesis of melanin, which protects against harmful rays from the sun. In this sense, inhibiting tyrosinase can control hyperpigmentation problems [38]. Overall, several compounds have been synthesized as enzyme inhibitors, but most of them have unpleasant side effects such as toxicity and gastrointestinal disorders. From this perspective, synthetic inhibitors must replace safe and effective natural inhibitors.

In the context of the enzyme inhibitory assay, the aqueous extracts from both bark and leaf sources demonstrated the most substantial inhibition of acetylcholinesterase (AChE), with values of 2.41 mg GALAE/g and 2.25 mg GALAE/g, respectively (Table 4). In contrast, the infusion extracts from both plant parts exhibited reduced AChE inhibition, measuring 0.33 mg GALAE/g for leaves and 0.70 mg GALAE/g for bark extracts. The inhibitory activity of these extracts displayed variations when tested against BChE, suggesting potential differences in their pharmacological profiles and highlighting the relevance of further exploration in this regard. The ethanolic (80%) extracts consistently demonstrated superior inhibition of various enzymes, including BchE, tyrosinase, and glucosidase, when compared to the other extracts. The inhibition of BchE by 80% ethanolic extracts derived from both leaves and barks exhibited a significantly enhanced and comparable enzyme inhibition rate, quantified at 3.08 GALAE/g. Subsequently, the n-hexane extract from stem barks displayed an inhibition rate of 2.31 mg GALAE/g. This outcome exhibited a direct correlation with the total phenolic and flavonoid content of these extracts. Prominent among the constituents of the 80% ethanolic extracts are significant flavonoids, including derivatives of isorhamnetin, quercetin, (epi)catechin, and apigenin. These flavonoids represent a vital component of the extract’s chemical composition, contributing to its potential health-related properties. In support of our findings, Temel et al. conducted a comprehensive study in which they reported that quercetin, a prominent flavonoid found in the 80% ethanolic extracts, plays a noteworthy role in the inhibition of BchE [39]. Their research reinforces the significance of quercetin and its potential therapeutic implications, particularly in the context of enzyme inhibition, further underlining the value of these flavonoid-rich extracts. Similarly, 80% ethanolic extract exhibited superior tyrosinase inhibition, with 64.89 mg KAE/g for the stem bark extract and 59.89 mg KAE/g for the leaves extract. This was followed by the n-hexane extract of stem barks at 47.01 mg KAE/g and the leaves extract at 43.43 mg KAE/g. The aqueous and infusion extracts from both parts of the tested plant demonstrated no significant inhibition of tyrosinase. Flavonoids primarily consist of phenolic rings and possess potent antioxidant properties. A study has substantiated the tyrosinase inhibitory capacity of isorhamnetin, a flavone [40,41]. The inhibitory efficacy of isorhamnetin is strongly associated with its polyphenolic constituents, as affirmed by a docking study [41]. In terms of the glucose digestive enzyme amylase, all the extracts exhibited decreased and nearly identical levels of inhibition. However, when examining the glucosidase enzyme, the 80% ethanolic stem bark extract displayed the highest level of inhibition (1.26 mmol ACAE/g), closely followed by the 80% ethanolic leaves extract (1.19 mmol ACAE/g), and the n-hexane bark extract (1.08 mmol ACAE/g) (Table 4). Conversely, the infusion and aqueous extracts from both leaves and stem bark demonstrated notably lower levels of inhibition, ranging from 0.04 to 0.97 mmol ACAE/g. It is significant to highlight that this study represents the first reported instance of enzyme inhibitory activity by *Vepris boiviniana* extract for all five enzymes examined.

### 2.5. Cytotoxicity and Antiviral Properties

The lowest cytotoxicity on VERO cells was observed for *V. boiviniana* ethanolic leaf extracts, and based on the obtained CC_50_ value (613.27 µg/mL) and literature data [42], it can be concluded that this extract was not toxic to non-cancerous cells. However, the ethanolic leaf extracts showed selective cytotoxicity towards all cancer cells, with a selectivity index (SI) between 5.14 and 9.66 (Table 5), with H1HeLa cells being the most sensitive. Based on literature data [43,44], SI > 3 might indicate significant anticancer selectivity. The ethanolic extract from bark showed higher toxicity on VERO and simultaneously lower anticancer selectivity. The *V. boiviniana* bark infusion was also non-toxic to VERO cells and aqueous leaf extract and infusion showed low toxicity, without any noticeable anticancer selectivity, with the exception of leaves infusion (VLi), which exerted significant selectivity towards FaDu cells. The highest toxicity towards non-cancerous cells was found for hexane extracts, both from leaves and bark, with bark hexane extract showing noticeable selectivity towards H1HeLa and RKO cells.

Plants from the genus *Vepris* have already been reported to exert anticancer potential [45]; however, the cytotoxicity and anticancer activity of *V. boiviniana* were described herein for the first time. The ethanolic and petroleum extract of the stem bark of *V. grandifolia* (Syn. *Teclea grandifloria*) were shown to inhibit the KB cell line with ED_50_ values of 30 and 6.2 µg/mL, respectively [46]. The KB cell line is a subline of HeLa. *V. lanceolate* root ethyl acetate extract was also toxic to HT-29 (colorectal adenocarcinoma) and KB cells, with CC_50_ of 30 and 45 µg/mL, respectively, while the leaf extract was significantly less toxic [47]. Cytotoxicity of *V. glomerata* originating from Tanzania was tested towards RT-4 (urinary bladder, transitional cell papilloma), HT-29, and A413 (epidermoid carcinoma) cells [48], while the extracts from the leaves of *V. soyauxii* collected in Cameroon against a panel of multi-drug resistant cancer-derived cell lines, including drug-sensitive (CCRF-CEM) and multidrug-resistant (CEM/ADR5000) leukemia cells, breast cancer cells (sensitive—MDA-MB-231, and resistant—MDA-MB-231/BCRP), wild-type colon cancer cells (HCT116(p53+/+)) and their knockout clone (HCT116(p53−/−)), human glioblastoma multiforme (U87MG), and hepatoblastoma cell line (HepG2), showed low CC_50_ values, between 4.06 and 13.6 µg/mL [49]. The *V. boiviniana* ethanolic leaf extract showed no toxicity towards non-cancerous cells and simultaneously significant anticancer effects against all three tested neoplastic cell lines. Interestingly, this extract was also the only one containing isorhamnetin, which was previously described as a promising anticancer candidate. Isorhamnetin, in combination with classic autophagy/mitophagy inhibitor, was shown to be effective against triple-negative breast cancer, which accounts for 15–20% of diagnosed breast tumors [50]. Moreover, isorhamnetin showed anticancer activity against the colon [51], gallbladder [52], gastric [53], melanoma [54], ovarian [55], pancreatic [56], prostate [57], and skin [58] cancer-derived cell lines. The anticancer effect of isorhamnetin may be associated with mitochondria-dependent apoptosis [53], inactivation of PI3K/AKT signaling pathway [51,52,57], or S phase cell cycle arrest [55,56]. Isorhamnetin also sensitizes doxorubicin-resistant breast cancer cells to doxorubicin, which provides a novel alternative for the treatment of drug-resistant cancer [59]. Recently, isorhamnetin was shown to exert in vivo anti-tumor activity against N-diethylnitrosamine and carbon tetrachloride induced hepatocellular carcinoma in Swiss albino mice [60].

Incubation of virus-infected cells with *V. boiviniana* extracts did not inhibit the formation of CPE. An example of virus-induced CPE and the effect of selected extracts is shown in Figure 1. Subsequent viral load assessment showed that all extracts decreased the HHV-1 load by 0.14–0.54 log and the HRV-14 viral load by 0.13–0.72 log (Figure 2). This result indicates a lack of significant antiviral activity. Acyclovir used as a reference anti-herpesviral drug inhibited the replication of HHV-1, and the viral load could not be evaluated. There are no approved treatment options for HRV-14; however, ribavirin at 500 and 250 µg/mL managed to reduce the HRV-14 viral load by 3.29 and 2.11 log, respectively. Several species of *Vepris*, including *V. louisii*, *V. trichocarpa*, *V. afzelii*, and *V. leandriana*, were reported as having antibacterial or antifungal activities [45]. However, antiviral activities have not been evaluated before. 

## 3. Materials and Methods

### 3.1. Plant Material

The samples of *Vepris boiviniana* (Baill.) Mziray was collected in the village of M’Kazi (Region of Bambao-Comoros) in September 2020, and the botanist Dr. Andilyat Mohamed (Université des Comores, Moroni, Comoros) authenticated it. The leaves and stem bark were dried at room temperature for one week, and then they were ground using a laboratory mill. The powdered plant materials were stored dark conditions at 4 °C.

### 3.2. Sample Preparation

The n-hexane, ethanol-aqueous (80%), and aqueous extracts were obtained through the maceration method, wherein 10 g of plant material were mixed with 200 mL of each solvent and left to steep for 24 h at room temperature. Afterwards, the mixtures were filtered using Whatman 1 filter paper, and the solvents were removed using a rotary evaporator. As for the aqueous extract (infusion), 10 g of plant material were infused in 200 milliliters of boiled water for 15 min, followed by filtration and lyophilization. Subsequently, all extracts were stored at 4 °C until analysis.

### 3.3. Total Phenolics and Flavonoids Content

Total phenolic and flavonoid contents of the extracts were determined by colorimetric methods previously described by Acquaviva et al. [34] and Llorent-Martínez et al. [61], respectively. The Folin–Ciocalteu and AlCl_3_ assays were utilized to determine the total phenolic and flavonoid contents, respectively. 

### 3.4. Liquid Chromatography—Mass Spectrometry Analysis

Chromatographic separation was performed on the C18 Gemini^®^ column (3 µm i.d. with TMS end-capping, 110 Å, 100 × 2 mm) equipped with a guard column (Phenomenex Inc, Torrance, CA, USA), using Agilent 1200 Infinity HPLC (Agilent Technologies, Santa Clara, CA, USA). Compounds were eluted by the solvent system composed of water with 0.1% formic acid *v*/*v* (A) and acetonitrile with 0.1% formic acid (B) pumped in the following program: 0–60% B for 45 min., next 60–95% B for 1 min, and 95% B for 9 min, at a flow rate of 0.2 mL/min. In total, 10 μL of the sample were injected into the chromatographic column at 20 °C. 

Detection was performed using the Agilent 6530B QTOF system (Agilent Technologies, Santa Clara, CA, USA) in positive and negative ion modes at the collision energies of 10 and 30 eV. A 50–1700 m/z range was scanned, and two spectra per second were acquired. The other conditions were as follows: drying gas temp: 275 °C, drying gas flow: 10 L/min, sheath gas temp: 325 °C, sheath gas flow: 12 L/min; nebulizer pressure: 35 psig, capillary V (+): 4000 V, skimmer: 65 V, and fragmentor: 140 V. Compounds were tentatively identified based on their accurate masses and fragmentation patterns, also supported by available literature sources.

### 3.5. In Vitro Antioxidant Assays

To assess the antioxidant potential of the extracts, six complementary in vitro spectrophotometric tests were performed [62]. These included the ABTS and DPPH assays, which examine the antioxidants’ ability to neutralize free radicals, the FRAP and CUPRAC assays, which evaluate the extract’s reduction capabilities, and the metal chelating ability (MCA) and phosphomolybdenum (PBD) assays. The assays were performed following the previously published methodologies [62]. Each of these assays, except for MCA, was carried out using the Trolox standard. The comparison for MCA was determined in terms of the EDTA equivalent per gram of extract.

### 3.6. Enzyme Inhibitory Activity

Enzyme inhibitory assays were performed as previously described [63]. The acetylcholinesterase (AChE) and butyrylcholinesterase (BChE) inhibition comparison were made as mg galanthamine equivalents (GALAE)/g extract; tyrosinase inhibition as mg kojic acid equivalents (KAE)/g extract; α-amylase and α-glucosidase inhibition were compared as mmol acarbose equivalents (ACAE)/g extract. 

### 3.7. Cytotoxicity and Antiviral Assays

Cytotoxicity and antiviral activity were assayed based on the previously described methodology [64]. Cell lines included non-cancerous VERO (kidney fibroblasts; ATCC, CCL-81), and cancer-derived cell lines—FaDu (hypopharyngeal cancer; ATCC, HTB-43), H1HeLa (cervical adenocarcinoma; ATCC, CRL-1958), and RKO (colon cancer; ATCC, CRL-2577). The Human Herpesvirus type 1 (HHV-1; ATCC, VR-260) was propagated in VERO cells, and Human Rhinovirus type 14 (HRV-14; ATCC, VR-284) in the H1HeLa cell line. VERO cells were cultured using DMEM (Corning, Tewksbury, MA, USA), while other cell lines in MEM (Corning) were also cultured. Cell media were supplemented with antibiotics (Penicillin-Streptomycin Solution, Corning) and fetal bovine serum (FBS, Corning). Phosphate-buffered saline (PBS) and trypsin were bought from Corning, whereas MTT (3-(4,5-dimethylthiazol-2-yl)-2,5-diphenyltetrazolium bromide) and DMSO (dimethyl sulfoxide) were obtained from Sigma (Sigma-Aldrich, St. Louis, MO, USA). Incubation was carried out in a 5% CO_2_ atmosphere at 37 °C (CO_2_ incubator, Panasonic Healthcare Co., Tokyo, Japan). Stock solutions of hexane and ethanolic extracts were prepared by dissolving the extracts in cell culture grade DMSO (PanReac Applichem), while aqueous extracts and infusion were dissolved in PBS. Stock solutions of extracts were stored frozen (−23 °C) until used.

Cytotoxicity was tested using an MTT-based protocol following a previously described protocol [64]. Briefly, the cells were passaged into 96-well plates (Falcon, TC-treated, Corning) and, after overnight incubation, treated with serial dilutions of extract or fraction stock solutions for 72 h. Subsequently, MTT-enriched media was added, and after 3h incubation, the precipitated formazan crystals were dissolved with SDS/DMF/PBS (14% SDS, 36% DMF, 50% PBS) mixture, and absorbance was measured (540 and 620 nm) with the Synergy H1 Multi-Mode Microplate Reader (BioTek Instruments, Inc., Winooski, VT, USA) with Gen5 software (ver. 3.09.07; BioTek Instruments, Inc.). Data analysis was performed with GraphPad Prism software (version 9.0, GraphPad Software, Inc., La Jolla, CA, USA), and the CC_50_ values (50% cytotoxic concentration) were evaluated. Additionally, for cancer cells, the selectivity indexes (SI) were assessed in relation to VERO (SI = CC_50_VERO/CC_50_Cancer, SI > 1 indicates anticancer selectivity). Data were statistically analyzed using GraphPad Prism (two-way ANOVA, Dunnett’s multiple comparisons test) and statistical differences between CC_50_ values on cancer and normal cells were evaluated.

The VERO or H1HeLa cells seeded in 48-well plates were infected (100-fold CCID_50_/mL; CCID_50_—50% cell culture infectious dose) with HHV-1 or HRV-14 for antiviral assays, leaving at least two uninfected wells as cell control. After 1h incubation, the virus-containing media were removed, cells were washed with PBS to remove unattached viral particles, and the extracts in non-toxic concentrations were added. Non-toxic concentrations were selected based on the results of cytotoxicity testing, as the concentrations did not decrease cellular viability by more than 10%. The experiment was conducted until the cytopathic effect (CPE) was observed (inverted microscope CKX41, Olympus Corporation, Tokyo, Japan) in the virus control (infected, non-treated cells). Afterwards, the infected cells treated with extracts were observed for possible CPE inhibition. Lastly, the plates were thrice frozen (−72 °C) and thawed; the samples were collected and stored at −72 °C until DNA or RNA isolation. Acyclovir and ribavirin were used as reference antiviral drugs. 

DNA isolation was carried out using a commercially available kit for DNA (QIAamp DNA Mini Kit, Cat#51304, QIAGEN GmbH, Hilden, Germany) or RNA (QIAamp RNA Mini Kit, Cat#52904, QIAGEN GmbH) following the manufacturer’s instructions. Real-time PCR amplification of HHV-1 DNA was performed using SsoAdvanced Universal SYBR Green Supermix (Bio-Rad Laboratories, Inc., Hercules, CA, USA) and primers (UL54F—5′CGCCAAGAAAATTTCATCGAG 3′, UL54R—5′ ACATCTTGCACCACGCCAG 3′) on the CFX96 thermal cycler (Bio-Rad Laboratories, Inc.). The amplification cycle parameters were as follows: initial activation (95 °C, 2 min); cycling (40 repeats: denaturation (95 °C, 10 s), annealing and synthesis (60 °C, 30 s), fluorescence acquisition); melting curve analysis (65–95 °C). For the assessment of HRV-14 RNA, the reverse transcription of viral RNA and subsequent amplification of the obtained cDNA (RT-qPCR) with enterovirus-specific primers (entrinR (5′-GAAACACGGACACCCAAAGTA-3′) and entrinF (5′-CGGCCCCTGAATGCGGCTAA-3′)) was performed using iTaq Universal SYBR Green One-Step Kit (Bio-Rad Laboratories, Inc.). The RT-qPCR parameters were as follows: reverse-transcription (50 °C, 10 min), activation of hot-start polymerase (95 °C, 1 min); cycling (40 repeats: denaturation (95 °C, 10 s), annealing and synthesis (60 °C, 30 s), fluorescence acquisition); melting curve analysis (65–95 °C). The reduction in HHV-1 or HRV-14 viral load in the tested samples was assessed in relation to virus control based on the relative quantity (ΔCq) method using CFX Manager™ Dx Software (version 3.1, Bio-Rad Laboratories).

### 3.8. Statistical Analysis

Statistical analysis was performed using Xl Stat (Version 16). All analyses were conducted in triplicates (n=3) and presented as mean values with their standard deviation (mean value ± std). Differences between samples were examined using one-way analysis of variance (ANOVA) and Tukey’s post hoc test with a significance level set at *p* < 0.05.

## 4. Conclusions

The study’s findings shed light on the distinct profiles of phenolic compounds and flavonoids present in various extracts prepared from the leaves and stem barks of *Vepris boiviniana*. Notably, we observed a higher concentration of phenolics in the leaf extract, whereas flavonoids were more abundant in the bark extract. Remarkably, we tentatively identified a total of 60 bioactive compounds across all tested extracts. The interesting result was the high antioxidant activity exhibited by the 80% ethanolic stem bark extract in the ABTS assay, indicating its potent antioxidant properties. Consistently, the infusion extract from the stem bark demonstrated higher antioxidant activity across all tested assays. Furthermore, the enzyme inhibitory assays revealed noteworthy findings. Aqueous extracts from both bark and leaf sources displayed substantial inhibition of AChE, while the ethanolic (80%) extract consistently demonstrated superior inhibition of various enzymes, including BChE, tyrosinase, and glucosidase, compared to the other extracts. Interestingly, the 80% ethanolic leaf extract exhibited the lowest cytotoxicity in VERO (non-cancer) cells and displayed selective cytotoxicity against cancer cells, particularly H1HeLa cells. On the other hand, the 80% ethanolic bark extract exhibited heightened toxicity in VERO cells but reduced selectivity against anticancer activity. Notably, the n-hexane extracts, especially from the leaves, displayed the highest toxicity towards non-cancerous cells while maintaining selectivity towards H1HeLa and RKO cells. In the context of viral load assessment, all extracts exhibited some reduction in HHV-1 and HRV-14 viral loads, although the reductions were limited. Briefly, our research highlights the diverse bioactive properties of *Vepris boiviniana* extracts, including potent antioxidant, enzyme inhibitory, and cytotoxicity against cancer cells. Nevertheless, further investigations are necessary to comprehensively evaluate the mechanisms underlying its biological activity and safety. Lastly, our work underscores the significance of preserving local knowledge related to the biodiversity of this plant. It serves as a reminder of the importance of conserving and respecting traditional wisdom in the context of medicinal plant utilization.

## Figures and Tables

**Figure 1 molecules-28-07531-f001:**
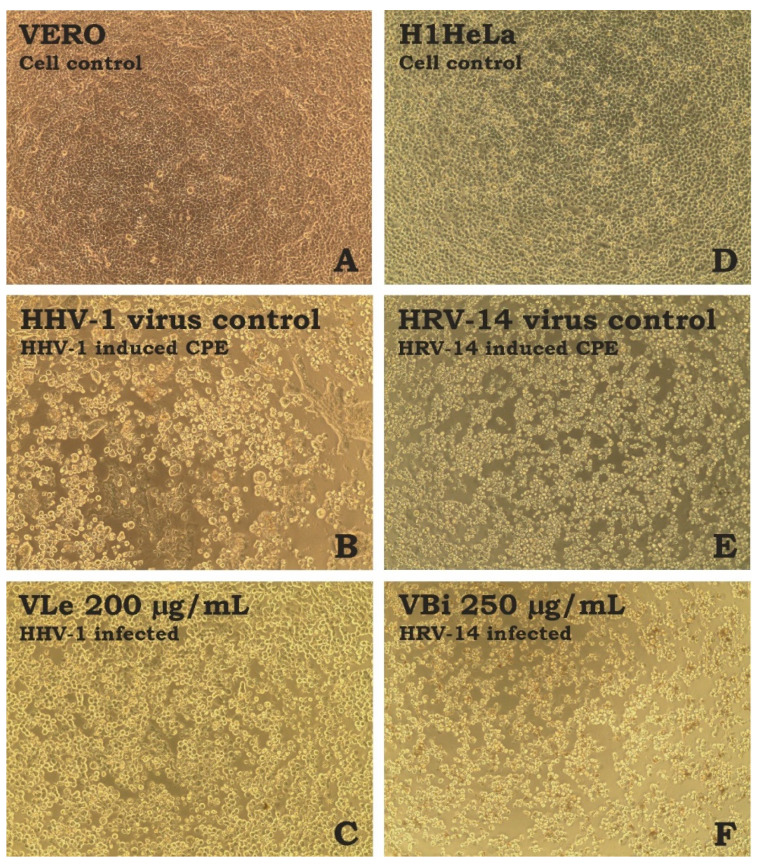
Influence of tested extracts on the virus-induced CPE ((**A**)—VERO cell control; (**B**)—HHV-1-induced CPE in VERO cells; (**C**)—HHV-1-infected VERO cells treated with VLe 200 µg/mL; (**D**)—H1HeLa cell control; (**E**)—HRV-14-induced CPE in H1HeLa cells; (**F**)—HRV-14-infected VERO cells treated with VBi 250 µg/mL; CPE—cytopathic effect).

**Figure 2 molecules-28-07531-f002:**
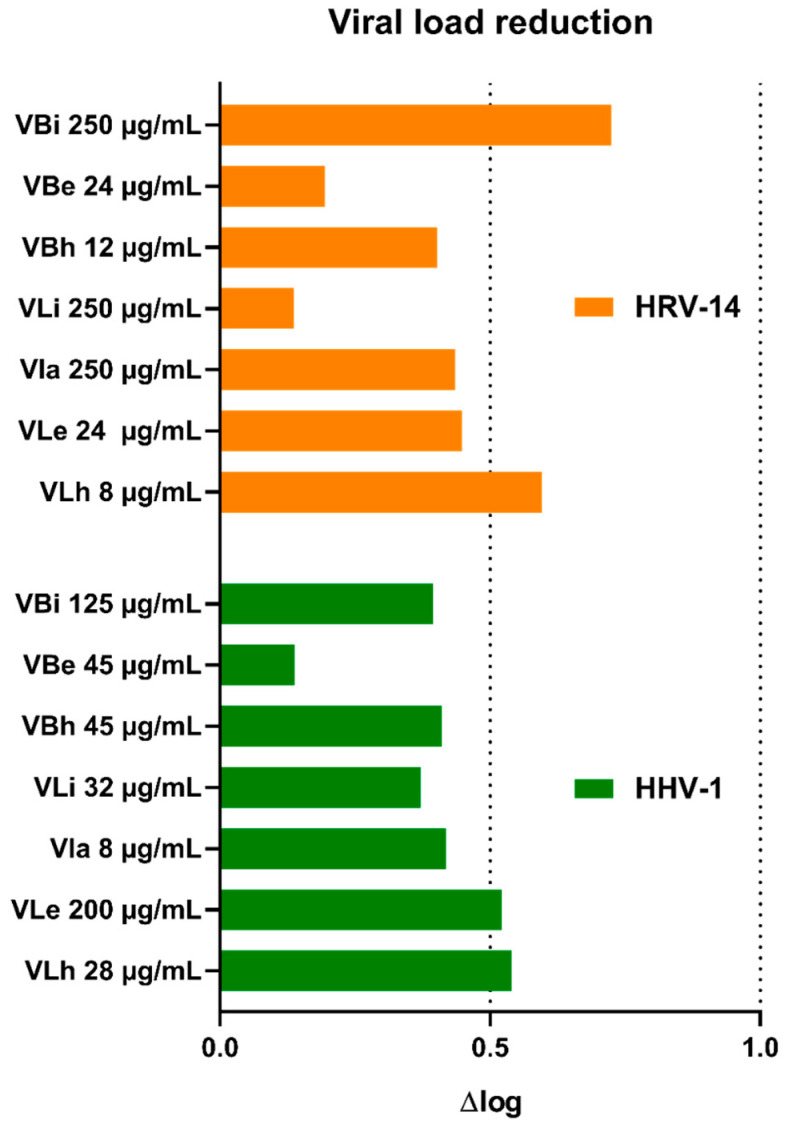
Reduction in viral load of HHV-1 and HRV-14 by *V. boiviniana* extracts (VB—*V. boiviniana* bark extracts; a—aqueous; e—ethanolic; h—hexane; i—infusion).

**Table 1 molecules-28-07531-t001:** Chemical composition of the tested extracts.

No	t_R_	Molecular Formula	Negative Ion Mode			Positive Ion Mode			Tentative Idenificaction	Extracts	References
			Precursor Ion Measured (*m*/*z*) (Δ, ppm)	Predicted (*m*/*z*)	Fragment Ions (*m*/*z*)	Precursor Ion (*m*/*z*) Measured (Δ, ppm)	Predicted (*m*/*z*)	Fragment Ions (*m*/*z*)			
1	1.58	C_6_H_14_O_6_	[M − H]^−^ 181.0718 (−0.21)	181.0718	59.0140; 89.0241; 101.0242; 71.0137; 163.0606; 73.0291	[M + Na]^+^ 205.0679 (1.97)	205.0683	82.9994; 55.0567; 82.0674; 84.0021; 97.0191; 56.0512	Hexitol	VLe, VLw, VLi, VBh, VBe, VBw, VBi	PubChem
2	1.75	C_5_H_10_O_6_	[M − H]^−^ 165.0399 (3.38)	165.0405	75.0089; 129.0188; 59.0142; 147.0300; 55.0201; 87.0093	-	-	-	Tetrahydroxypentanoic acid (Arabinoic acid)	VLe, VLw, VLi, VBe, VBw, VBi	PubChem
3	1.84	C_6_H_12_O_7_	[M − H]^−^ 195.0512 (−0.89)	195.0510	75.0099; 129.0195; 99.0091; 59.0151; 87.0089; 177.0406	-	-	-	Gluconic acid	VLe, VLw, VLi, VBe, VBw, VBi	PubChem
4	1.91	C_7_H_12_O_6_	[M − H]^−^ 191.0559 (1.10)	191.0561	85.0295; 59.0139; 93.0342; 127.0389; 87.0089; 75.0088	[M + Na]^+^ 215.0524 (1.09)	215.0526	172.0942; 154.0841; 85.0287; 99.0435; 111.0217; 197.0415	Quinic acid	VLe, VLw, VLi, VBh, VBe, VBw, VBi	PubChem
5	2.18	C_4_H_6_O_5_	[M − H]^−^ 133.0138 (3.33)	133.0142	115.0028; 71.0139; 72.9934; 89.0234; 116.0060	-	-	-	Malic acid	VLe, VLw, VLi, VBh, VBe, VBw, VBi	PubChem
6	2.26	C_6_H_8_O_7_	[M − H]^−^ 191.0196 (0.66)	191.0197	111.0081; 173.0082; 85.0294; 154.9977; 117.0186; 72.9933	[M + Na]^+^ 215.0163 (−0.40)	215.0162	172.0999; 101.0275; 154.0913; 83.0177; 129.0218; 197.0082	Citric acid	VLe, VLw, VLi, VBh, VBe, VBw, VBi	PubChem
7	4.92	C_14_H_26_O_10_	[M − COOH]^−^ 399.1502 (1.69)	399.1508	353.1451; 207.0873; 354.1470; 101.0258; 161.0457; 59.0139	[M + Na]^+^ 377.1420 (−0.51)	377.1418	71.0482; 85.0289; 247.0460; 101.9770; 155.9456; 170.9016	Carbohydrate	VBi, VBw, VBe	PubChem
8	10.59	C_8_H_10_O_3_	[M − H]^−^ 153.0560 (−1.83)	153.0557	123.0454; 124.0489; 110.9839; 122.0364; 95.0486	-	-	-	Hydroxytyrosol	VLw	PubChem
9	10.76	C_16_H_30_O_11_	[M − COOH]^−^ 443.1774 (−0.97)	443.1770	397.1731; 251.1137; 101.0252; 398.1769; 161.0423; 59.0151	[M + Na]^+^ 421.1682 (−0.42)	421.1680	275.1098; 177.0557; 276.1119; 197.0794; 165.0535; 121.0648	Carbohydrate	VBi, VBw, VBe	PubChem
10	10.76	C_16_H_30_O_9_	[M − COOH]^−^ 411.1878 (−1.68)	411.1872	365.1816; 89.0246; 366.1882; 119.0359; 203.1278; 59.0145	[M + Na]^+^ 389.1773 (2.47)	389.1782	203.0502; 209.1096; 371.1648; 227.1205; 163.0362; 230.1345	Terpene glycoside	VLe, VLw, VLi	PubChem
11	11.43	C_14_H_20_O_7_	[M − COOH]^−^ 345.1190 (0.35)	345.1191	137.0598; 299.1120; 161.0444; 101.0229; 71.0128; 300.1167	[M + Na]^+^ 323.1106 (−1.59)	323.1101	85.0298; 69.0340; 167.0163; 121.0637; 85.0633; 203.0536	Salidroside isomer	VLe, VLw, VLi, VBe, VBi	fragmentation PubChem
12	11.51	C_16_H_30_O_11_	[M − COOH]^−^ 443.1779 (−2.22)	443.1770	397.1708; 251.1151; 398.1752; 101.0257; 59.0149; 71.0133	[M + Na]^+^ 421.1684 (−0.92)	421.1680	275.1136; 276.1106; 121.0518; 117.9783; 167.0628; 205.0273	Carbohydrate	VBi, VBw	PubChem
13	11.84	C_7_H_6_O_4_	[M − H]^−^ 153.0194 (−0.44)	153.0193	109.0287; 108.0222; 66.9944; 81.0323; 53.0410; 91.0199	-	-	-	Dihydroxybenzoic acid	VLw, VLi, VBw, VBi	PubChem
14	12.18	C_14_H_20_O_8_	[M − H]^−^ 315.1086 (−0.19)	315.1085	123.0446; 153.0552; 124.0473; 154.0587; 122.0364; 109.0279	-	-	-	Hydroxythyrosol glucoside	VLe, VLw, VLi, VBe, VBi	fragmentation, PubChem
15	12.93	C_19_H_34_O_9_	[M − COOH]^−^ 451.2188 (−0.77)	451.2185	405.2145; 89.0246; 179.0546; 119.0338; 59.0150; 71.0129	[M + Na]^+^ 429.2080 (3.70)	429.2095	268.0942; 249.1412; 201.0003; 203.0486; 158.9949; 85.0250	Norisoprenoid glucoside derivative	VLe, VLw, VLi	fragmentation, PubChem
16	13.35	C_16_H_30_O_9_	[M − COOH]^−^ 411.1875 (−0.86)	411.1872	89.0239; 179.0565; 59.0143; 365.1825; 119.0342; 71.0139	[M + Na]^+^ 389.1772 (2.74)	389.1782	135.0390; 227.1181; 359.1620; 147.0376; 203.0461; 85.0241	Terpene glycoside	VLe, VLw, VLi	PubChem
17	14.02	C_7_H_6_O_3_	[M − H]^−^ 137.0244 (0.13)	137.0244	108.0212; 136.0161; 92.0266; 91.0188; 53.0407; 109.0270	-	-	-	Hydroxybenzoic acid isomer	VLe, VLw, VLi, VBe, VBw, VBi	PubChem
18	14.35	C_19_H_34_O_9_	[M − COOH]^−^ 451.2181 (0.95)	451.2185	405.2111; 89.0240; 59.0143; 406.2163; 101.0232; 71.0146	[M + Na]^+^ 429.2073 (5.43)	429.2095	98.9728; 201.0012; 158.9929; 411.1956; 85.0265; 203.0539	Norisoprenoid glucoside derivative	VLe, VLw, VLi	fragmentation, PubChem
19	14.86	C_18_H_19_NO_4_	[M − H]^−^ 312.1245 (−1.18)	312.1241	282.0769; 297.1001; 254.0812; 239.0699; 283.0788; 224.0481	[M + H]^+^ 314.1384 (0.91)	314.1387	222.0662; 191.0842; 282.0879; 265.0848; 237.0896; 219.0792	Alkaloid	VLe, VLi, VBh, VBe, VBw, VBi	PubChem
20	14.93	C_16_H_18_O_9_	[M − H]^−^ 353.0878 (0.02)	353.0878	191.0548; 179.0343; 135.0447; 192.0587; 85.0297; 209.0292	[M + H]^+^ 355.1025 (−0.40)	355.1024	-	Caffeoylquinic acid (3-*O*-caffeoylquinic acid)	VLi, VBe, VBi	PubChem [11,12,13]
21	15.35	C_7_H_6_O_3_	[M − H]^−^ 137.0244 (0.13)	137.0244	93.0344; 65.0402; 94.0387; 66.0429; 75.0254	[M + H]^+^ 139.0391 (−0.94)	139.0390	-	Hydroxybenzoic acid isomer	VLw, VBw	PubChem
22	15.36	C_19_H_21_NO_4_	[M − H]^−^ 326.1400 (−0.67)	326.1398	311.1156; 268.0850; 296.0925; 239.0699; 281.0683; 224.0475	[M + H]^+^ 328.1545 (−0.51)	328.1543	297.1113; 265.0856; 298.1150; 266.0887; 282.0879; 237.0903	Alkaloid	VLe, VLi, VBh, VBe, VBw, VBi	PubChem
23	15.52	C_19_H_34_O_9_	[M − COOH]^−^ 451.2194 (−2.25)	451.2185	405.2152; 89.0239; 59.0138; 119.0331; 179.0534; 406.2163	[M + Na]^+^ 429.2083 (2.96)	429.2095	249.1440; 209.1512; 147.0774; 99.0438; 85.0273; 152.0693	Norisoprenoid glucoside derivative	VLe, VLw, VLi	fragmentation, PubChem
24	16.52	C_30_H_26_O_12_	[M − H]^−^ 577.1350 (0.26)	577.1351	125.0252; 289.0689; 407.0742; 245.0817; 151.0402; 161.0276	[M + H]^+^ 579.1486 (1.91)	579.1497	289.0668; 291.0834; 427.1004; 127.0366; 409.0903; 247.0570	Procyanidin dimer type B	VBh, VBe, VBi	fragmentation, PubChem, [14]
25	17.43	C_15_H_14_O_6_	[M − H]^−^ 289.0717 (0.21)	289.0718	245.0821; 109.0287; 203.0711; 125.0234; 205.0503; 151.0398	[M + H]^+^ 291.0850 (4.53)	291.0863	139.0366; 123.0418; 165.0522; 147.0417; 140.0398; 207.0619	Epicatechin	VBh, VBe, VBi	fragmentation, PubChem, [14,15]
26	17.77	C_16_H_18_O_9_	[M − H]^−^ 353.0879 (−0.27)	353.0878	191.0558; 192.0589; 85.0295; 193.0594; 209.0293; 161.0235	[M + H]^+^ 355.1020 (1.01)	355.1024	-	Caffeoylquinic acid (5-*O*-caffeoylquinic acid)	VLe, VLw, VLi, VBh, VBe, VBi	PubChem, [11,12,13]
27	18.27	C_16_H_18_O_9_	[M − H]^−^ 353.0879 (−0.27)	353.0878	191.0550; 173.0449; 179.0340; 135.0440; 85.0296; 192.0578	-	-	-	Caffeoylquinic acid (4-*O*-caffeoylquinic acid)	VLi, VBe, VBi	PubChem, [11,12,13]
28	18.35	C_9_H_8_O_4_	[M − H]^−^ 179.0355 (−2.88)	179.0350	135.0448; 134.0372; 89.0403; 107.0504; 79.0546; 136.0468	-	-	-	Caffeic acid	VLw, VBi	[16,17,18]
29	18.44	C_33_H_40_O_21_	[M − H]^−^ 771.1985 (0.56)	771.1989	299.0210; 301.0362; 300.0278; 609.1506; 462.0849; 463.0881	[M + H]^+^ 773.2124 (1.40)	773.2135	465.1007; 627.1523; 303.0478; 466.1045; 611.1585; 628.1545	Pentahydroxyflavone *O*-rutinoside *O*-hexoside (Quercetin 3-*O*-rutinoside-7-*O*-glucoside)	VLe, VLi	fragmentation, PubChem
30	18.77	C_17_H_32_O_10_	[M − COOH]^−^ 441.1982 (−1.14)	441.1978	395.1916; 249.1333; 396.1962; 101.0240; 71.0145; 59.0136	[M + Na]^+^ 419.1884 (0.93)	419.1888	-	Carbohydrate	VBi, VBw, VBe, VLi, VLw, VLe	PubChem
31	18.95	C_30_H_26_O_11_	[M − H]^−^ 561.1423 (−3.67)	561.1402	289.0682; 165.0551; 125.0281; 435.1033; 407.0783;	-	-	-	Procyanidin dimer ((epi)Catechin-(epi)Afzelechin)	VBe	PubChem, [19]
32	19.53	C_33_H_40_O_20_	[M − H]^−^ 755.2034 (0.82)	755.2040	285.0402; 593.1524; 283.0235; 284.0314; 594.1574; 255.0276	[M + H]^+^ 757.2195 (0.22)	757.2197	-	Tetrahydroxyflavone *O*-rutinoside *O*-hexoside (Kaempferol 3-*O*-rutinoside-7-*O*-glucoside)	VLe, VLi	Fragmentation, PubChem
33	19.86	C_34_H_42_O_21_	[M − H]^−^ 785.2160 (−1.80)	785.2146	315.0504; 623.1632; 314.0402; 300.0305; 299.0178; 271.0259	[M + H]^+^ 787.2292 (−0.08)	787.2291	-	Tetrahydroxymethoxyflavone *O*-rutinoside *O*-hexoside (Isorhamnetin 3-*O*-rutinoside-glucoside)	VLe, VLi	Fragmentation PubChem
34	20.19	C_45_H_36_O_18_	[M − H]^−^ 863.1821 (0.91)	863.1829	411.0743; 289.0724; 285.0425; 711.1369; 451.1040; 412.0785	[M + H]^+^ 865.1972 (0.28)	865.1974	533.1065; 713.1485; 287.0526; 695.1387; 575.1167; 739.1642	Procyanidin trimer	VLe, VLi, VBh, VBe, VBi	Fragmentation, PubChem
35	20.22	C_16_H_18_O_8_	[M − H]^−^ 337.0934 (−1.51)	337.0929	191.0563; 93.0353; 119.0508; 87.0092; 85.0299; 163.0404	-	-	-	Coumaroylquinic acid isomer	VLe, VLi, VBe, VBi	Fragmentation, PubChem, [11,12,13]
36	20.95	C_39_H_32_O_15_	[M − H]^−^ 739.1657 (1.55)	739.1668	289.0721; 177.0197; 339.0505; 587.1186; 449.0865; 290.0707	[M + H]^+^ 741.1817 (−0.41)	741.1814	-	Tannin	VBe, VBi	Fragmentation, PubChem
37	21.19	C_16_H_18_O_8_	[M − H]^−^ 337.0925 (1.16)	337.0929	191.0547; 85.0299; 192.0574; 93.0342; 127.0398; 59.0149	-	-	-	Coumaroylquinic acid isomer	VLe, VLi	Fragmentation, PubChem, [11,12,13]
38	22.28	C_32_H_38_O_20_	[M − H]^−^ 741.1896 (−1.66)	741.1884	300.0286; 301.0308; 271.0255; 742.1901; 255.0309; 178.9962	[M + H]^+^ 743.2028 (1.64)	743.2040	303.0475; 304.0508; 465.1008; 611.1563; 743.2041; 85.0275	Pentahydroxyflavone *O*-rutinoside-pentoside (Quercetin 3-*O*-rutinoside-pentoside)	VLe, VLw, VLi	Fragmentation, PubChem
39	22.52	C_10_H_10_O_4_	[M − H]^−^ 193.0509 (−1.38)	193.0506	134.0375; 178.0270; 149.0610; 93.0348; 137.0228; 135.0433	-	-	-	Isoferulic acid	VLw	PubChem
40	22.70	C_18_H_34_O_10_	[M − COOH]^−^ 455.2145 (−2.68)	455.2134	409.2081; 263.1491; 410.2107; 101.0243; 264.1529; 411.2163	[M + Na]^+^ 433.2047 (−0.69)	433.2044	-	Hexyl 2-*O*-t(6-Deoxy-Alpha-L-Galactopyranosyl)-Beta-D-Galactopyranoside	VLe, VLw, VLi, VBe, VBw, VBi	PubChem
41	23.19	C_26_H_28_O_16_	[M − H]^−^ 595.1312 (−1.24)	595.1305	300.0288; 271.0236; 301.0319; 255.0282; 272.0292; 302.0331	[M + H]^+^ 597.1458 (−1.32)	597.1450	-	Pentahydroxyflavone *O*-hexoside *O*-pentoside (Quercetin *O*-glucoside *O*-pentoside)	VLe, VLw, VLi	Fragmentation, PubChem
42	23.45	C_27_H_30_O_16_	[M − H]^−^ 609.1453 (1.33)	609.1461	300.0265; 301.0334; 271.0239; 255.0279; 151.0025; 243.0297	[M + H]^+^ 611.1608 (−0.23)	611.1607	303.0487; 465.1022; 85.0289; 129.0542; 147.0646	Rutin	VLe, VLw, VLi, VBe, VBi	Fragmentation, PubChem
43	23.70	C_15_H_12_O_7_	[M − H]^−^ 303.0513 (−0.90)	303.0510	285.0401; 125.0238; 177.0200; 286.0451; 175.0402; 151.0032	-	-	-	Pentahydroxydihydroflavone (Taxifolin)	VBe, VBi	Fragmentation, PubChem, [20]
44	24.19	C_21_H_20_O_12_	[M − H]^−^ 463.0889 (−1.51)	463.0882	300.0261; 271.0239; 301.0323; 255.0288; 243.0277; 151.0023	[M + H]^+^ 465.1023 (0.98)	465.1028	303.0491; 85.0290; 61.0298; 97.0284; 91.0394; 73.0287	Pentahydroxyflavone *O*-hexoside (Quercetin 3-*O*-glucoside)	VLe, VLw, VLi, VBe, VBi	Fragmentation, PubChem
45	24.78	C_9_H_16_O_4_	[M − H]^−^ 187.0978 (−1.16)	187.0976	125.0971; 97.0663; 126.0998; 169.0864; 123.0799; 143.1070	-	-	-	Azelaic acid	VLe, VLw, VLi, VBh, VBe, VBw, VBi	PubChem
46	24.95	C_27_H_30_O_15_	[M − H]^−^ 593.1532 (−3.38)	593.1512	285.0401; 284.0329; 255.0302; 227.0356; 286.0441; 229.0504	[M + H]^+^ 595.1659 (−0.26)	595.1657	287.0534; 449.1074; 288.0573; 450.1108; 85.0277; 129.0532	Tetrahydroxyflavone *O*-rutinoside (Kaempferol 3-*O*-rutinoside = Nicotiflorin)	VLe, VLw, VLi	Fragmentation, PubChem
47	25.11	C_28_H_32_O_16_	[M − H]^−^ 623.1636 (−2.95)	623.1618	315.0483; 300.0240; 314.0411; 299.0182; 271.0225; 243.0284	[M + H]^+^ 625.1754 (1.46)	625.1763	317.0601; 318.0645; 479.1136; 480.1169; 85.0259; 129.0510	Tetrahydroxymethoxyflavone *O*-rutinoside (Isorhamnetin 3-*O*-rutinoside)	VLe, VLw, VLi	Fragmentation, PubChem
48	25.28	C_20_H_18_O_11_	[M − H]^−^ 433.0787 (−2.45)	433.0776	300.0258; 271.0255; 255.0293; 301.0340; 243.0294; 151.0036	[M + H]^+^ 435.0929 (−1.64)	435.0922	303.0465; 304.0503; 73.0267; 61.0270; 229.0453; 165.0153	Pentahydroxyflavone *O*- pentoside (Quercetin *O*-pentoside)	VLe, VLw, VLi, VBi	Fragmentation
49	25.61	C_21_H_18_O_13_	[M − H]^−^ 477.0685 (−2.17)	477.0675	301.0362; 151.0033; 178.9972; 121.0292; 107.0118; 255.0337	-	-	-	Pentahydroxyflavone *O*- glucuronide (Quercetin 3-*O*-glucuronide)	VBe, VBi	Fragmentation, PubChem
50	25.78	C_21_H_20_O_11_	[M − H]^−^ 447.0941 (−1.82)	447.0933	300.0270; 271.0244; 301.0337; 255.0298; 151.0033; 243.0307	[M + H]^+^ 449.1073 (1.20)	449.1078	303.0483; 287.0533; 85.0284; 71.0493; 57.0337; 129.0531	Pentahydroxyflavone *O*-rhamnoside (Quercetin 3-*O*-rhamnoside)	VLe, VLw, VLi, VBh, VBe, VBi	Fragmentation, PubChem
51	25.94	C_22_H_22_O_12_	[M − H]^−^ 477.1030 (1.78)	477.1038	314.0425; 243.0285; 271.0235; 285.0397; 257.0432; 299.0185	[M + H]^+^ 479.1174 (2.10)	479.1184	317.0634; 318.0662; 85.0262; 302.0393; 97.0263; 285.0365	Tetrahydroxymethoxyflavone *O*-hexoside (Isorhamnetin 3-*O*-glucoside)	VLe, VLw, VLi	Fragmentation, PubChem
52	26.04	C_25_H_24_O_12_	[M − H]^−^ 515.1207 (−2.33)	515.1195	353.0877; 191.0570; 179.0355; 354.0913; 135.0454; 173.0442	-	-	-	*di*-Caffeoylquinic acid isomer	VLi, VBh, VBe, VBi	PubChem, [11,13]
53	26.37	C_21_H_36_O_10_	[M − COOH]^−^ 493.2304 (−3.01)	493.2291	447.2228; 448.2254; 315.1811; 101.0241; 71.0134; 161.0448	[M + Na]^+^ 471.2198 (0.60)	471.2201	335.0914; 471.2188; 336.0947; 472.2213; 337.0962; 275.0689	Terpene glycoside	VLe, VLw, VLi, VBh, VBe, VBw, VBi	PubChem
54	27.04	C_25_H_24_O_12_	[M − H]^−^ 515.1202 (−1.36)	515.1195	353.0872; 173.0445; 179.0349; 191.0551; 354.0892; 135.0442	-	-	-	*di*-Caffeoylquinic acid isomer	VLi, VBh, VBe, VBi	PubChem, [11,13]
55	27.37	C_30_H_26_O_15_	[M − H]^−^ 625.1199 (−0.01)	625.1199	300.0268; 301.0320; 463.0909; 271.0202; 464.0909; 255.0309	-	-	-	Pentahydroxyflavone *O*-Caffeoyl-hexoside (Quercetin 3-*O*-caffeoyl-glucoside)	VBe	fragmentation, PubChem
56	27.70	C_21_H_20_O_10_	[M − H]^−^ 431.0993 (−2.15)	431.0984	255.0294; 227.0342; 284.0315; 285.0396; 256.0341; 228.0382	[M + H]^+^ 433.1122 (1.67)	433.1129	287.0526; 71.0476; 288.0560; 57.0323; 72.0504; 153.0142	Tetrahydroxyflavone *O*-rhamnoside (Kaempferol 3-*O*-rhamnoside)	VLe, VLi, VBe, VBi	fragmentation, PubChem
57	33.71	C_18_H_34_O_5_	[M − H]^−^ 329.2332 (0.45)	329.2333	171.1027; 211.1340; 229.1466; 139.1130; 212.1359; 99.0818	-	-	-	Fatty acid	VLe, VLw, VLi, VBh, VBe, VBw, VBi	PubChem
58	33.96	C_15_H_10_O_5_	[M − H]^−^ 269.0455 (0.17)	269.0455	117.0347; 65.0043; 151.0018; 107.0139; 118.0371; 149.0254	-	-	-	Trihydroxyflavone (Apigenin)	VBe	fragmentation, PubChem, [21]
59	34.46	C_16_H_12_O_7_	[M − H]^−^ 315.0511 (−0.23)	315.0510	300.0274; 151.0027; 63.0245; 107.0138; 108.0219; 83.0137	-	-	-	Tetrahydroxymethoxyflavone (Isorhamnetin)	VLe	fragmentation, PubChem
60	35.63	C_18_H_34_O_5_	[M − H]^−^ 329.2341 (−2.28)	329.2333	201.1164; 199.1327; 171.1060; 211.1342; 202.1133; 59.0147	-	-	-	Fatty acid	VLe, VLw, VLi, VBe, VBw, VBi	PubChem

tr—retention time; Vle—leaf ethanol (80%), Vlw—leaf water, Vli—leaf infusion, Vbh—bark n-hexane, Vbe—bark ethanol (80%), Vbw—bark water, Vbi—bark infusion.

**Table 2 molecules-28-07531-t002:** Total phenolic and flavonoid content of the tested extracts *.

Part	Extracts	Total Phenolic Content (mg GAE/g)	Total Flavonoid Content (mg RE/g)
Leaves	n-hexane	29.43 ± 3.27 ^g^	20.73 ± 0.58 ^b^
	Ethanol (80%)	70.61 ± 1.42 ^c^	25.39 ± 0.68 ^a^
	Water	34.98 ± 0.25 ^f^	13.12 ± 0.27 ^c^
	Infusion	46.44 ± 0.62 ^d^	25.72 ± 0.42 ^a^
Stem barks	n-hexane	27.49 ± 0.55 ^g^	3.42 ± 0.08 ^e^
	Ethanol (80%)	108.19 ± 0.98 ^a^	6.13 ± 0.27 ^d^
	Water	42.27 ± 0.68 ^e^	1.17 ± 0.38 ^f^
	Infusion	75.85 ± 0.35 ^b^	4.11 ± 0.18 ^e^

* Values are reported as mean ± SD of three parallel measurements. GAE: Gallic acid equivalents; RE: Rutin equivalents. Different letters indicate significant differences in the tested extracts (*p* < 0.05).

**Table 3 molecules-28-07531-t003:** Antioxidant properties of the tested extracts *.

Part	Extracts	DPPH (mg TE/g)	ABTS (mg TE/g)	CUPRAC (mg TE/g)	FRAP (mg TE/g)	PBD (mmol TE/g)	MCA (mg EDTAE/g)
Leaves	n-hexane	10.95 ± 0.34 ^e^	23.91 ± 0.60 ^g^	62.98 ± 3.01 ^g^	36.48 ± 0.53 ^f^	2.32 ± 0.06 ^a^	11.33 ± 0.64 ^d^
	Ethanol (80%)	132.61 ± 6.07 ^b^	203.34 ± 5.21 ^c^	200.09 ± 1.15 ^c^	122.57 ± 0.76 ^c^	1.93 ± 0.01 ^b^	9.92 ± 0.80 ^e^
	Water	39.33 ± 0.26 ^d^	85.64 ± 1.71 ^e^	83.99 ± 0.98 ^f^	65.72 ± 0.44 ^e^	0.88 ± 0.02 ^g^	25.37 ± 0.19 ^b^
	Infusion	59.06 ± 1.34 ^c^	133.29 ± 5.21 ^d^	115.92 ± 1.00 ^d^	76.42 ± 1.17 ^d^	1.08 ± 0.02 ^f^	28.43 ± 0.04 ^a^
Stem barks	n-hexane	30.21 ± 1.15 ^d^	51.26 ± 3.59 ^f^	65.47 ± 1.03 ^g^	36.78 ± 0.55 ^f^	1.49 ± 0.09 ^d^	18.31 ± 0.65 ^c^
	Ethanol (80%)	388.42 ± 9.96 ^a^	551.82 ± 9.20 ^a^	512.03 ± 2.39 ^a^	320.28 ± 7.02 ^a^	2.41 ± 0.09 ^a^	25.47 ± 0.26 ^b^
	Water	32.26 ± 1.89 ^d^	88.25 ± 0.17 ^e^	106.51 ± 0.25 ^e^	80.20 ± 1.17 ^d^	1.27 ± 0.05 ^e^	28.86 ± 0.54 ^a^
	Infusion	137.39 ± 4.20 ^b^	216.38 ± 3.85 ^b^	218.46 ± 0.70 ^b^	138.28 ± 2.22 ^b^	1.70 ± 0.07 ^c^	29.54 ± 0.12 ^a^

* Values are reported as mean ± SD of three parallel measurements. PBD: phosphomolybdenum; MCA: metal chelating activity; TE: Trolox equivalent; EDTAE: EDTA equivalent. Different letters indicate significant differences in the tested extracts (*p* < 0.05).

**Table 4 molecules-28-07531-t004:** Enzyme inhibitory effects of the tested extracts*.

Part	Extracts	AChE (mg GALAE/g)	BChE (mg GALAE/g)	Tyrosinase (mg KAE/g)	Amylase (mmol ACAE/g)	Glucosidase (mmol ACAE/g)
Leaves	n-hexane	1.69 ± 0.07 ^d^	1.62 ± 0.03 ^c d^	43.43 ± 0.82 ^d^	0.71 ± 0.03 ^a^	0.22 ± 0.06 ^e^
	Ethanol (80%)	1.67 ± 0.08 ^d^	3.08 ± 0.14 ^a^	59.89 ± 0.45 ^b^	0.40 ± 0.01 ^c^	1.19 ± 0.01 ^a^
	Water	2.25 ± 0.03 ^b^	1.32 ± 0.22 ^d^	na	0.06 ± 0.02 ^f^	0.37 ± 0.05 ^d^
	Infusion	0.70 ± 0.01 ^e^	1.75 ± 0.14 ^c^	na	0.05 ± 0.01 ^f^	0.04 ± 0.01 ^f^
Stem barks	n-hexane	0.66 ± 0.07 ^e^	2.31 ± 0.21 ^b^	47.01 ± 0.32 ^c^	0.49 ± 0.01 ^b^	1.08 ± 0.03 ^b^
	Ethanol (80%)	2.03 ± 0.17 ^c^	3.08 ± 0.06 ^a^	64.89 ± 0.40 ^a^	0.47 ± 0.01 ^b^	1.26 ± 0.01 ^a^
	Water	2.41 ± 0.01 ^a^	0.23 ± 0.01 ^e^	na	0.13 ± 0.01 ^d^	0.90 ± 0.01 ^c^
	Infusion	0.33 ± 0.02 ^f^	1.82 ± 0.08 ^c^	na	0.09 ± 0.01 ^e^	0.97 ± 0.01 ^c^

* Values are reported as mean ± SD of three parallel measurements. GALAE: Galantamine equivalent; KAE: Kojic acid equivalent; ACAE: Acarbose equivalent; na: not active. Different letters indicate significant differences in the tested extracts (*p* < 0.05).

**Table 5 molecules-28-07531-t005:** Cytotoxicity and anticancer selectivity of *V. boiviniana* extracts.

*Vepris boiviniania*	VERO	FaDu		H1HeLa		RKO	
	CC_50_	CC_50_	SI	CC_50_	SI	CC_50_	SI
Leaves—hexane (VLh)	63.06 ± 5.30	82.82 ± 9.53	0.76	26.20 ± 0.59 *	2.41	22.04 ± 3.24 *	2.86
Leaves—ethanol (VLe)	613.27 ± 62.25	114.90 ± 11.77	5.34 **	63.46 ± 2.03 **	9.66	119.33 ± 10.21 **	5.14
Leaves—aqueous (VLa)	251.4 ± 26.98	254.4 ± 20.52	0.99	>500	<1	>500	<1
Leaves—infusion (VLi)	332.8 ± 25.44	125.5 ± 13.33 **	2.65	>500	<1	>500	<1
Bark—hexane (VBh)	81.01 ± 11.24	56.50 ± 3.52	1.43	43.29 ± 4.06 *	1.87	49.55 ± 5.32 *	1.63
Bark—ethanol (VBe)	206.63 ± 7.71	118.47 ± 8.72 **	1.74	80.03 ± 11.59 **	2.58	102.45 ± 5.75 **	2.02
Bark—infusion (VBi)	>500	296.85 ± 21.71	>1.68	>500	na	306.20 ± 9.19	>1.63

CC_50_—50% cytotoxic concentration (mean ± SD; µg/mL); SI—selectivity index (CC_50_VERO/CC_50_CancerCells), na—not applicable; *—statistically significant (*p* < 0.05); **—statistically highly significant (*p* < 0.001). Statistical significance calculated with reference to VERO cells.

## Data Availability

Data are contained within the article.

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
