# Peer review of "Evaluating Phytochemical Profiles, Cytotoxicity, Antiviral Activity, Antioxidant Potential, and Enzyme Inhibition of Vepris boiviniana Extracts"

_molecules, 2023, doi:10.3390/molecules28227531_

Round 1
Reviewer 1 Report
Comments and Suggestions for Authors
The manuscript is presented clearly and consists of different analyses that appear to be essential in describing the characteristics of V. boiviniana as well as its potential as antioxidant and in cancer treatment. I have some comments for improvement:
1. Table 1: would you be able to add the information regarding the concentration of each molecule in the each extract? I believe it would be interesting for the readers.
2. Table 4: In the discussion, please add some information about the interest of choosing the tested enzymes. To which diseases or biological processes are those enzymes linked? What is the interest of inhibiting the activity of such enzymes?
3. Table 5: would it be possible to add statistical analysis to the data? Could you describe in the discussion the criteria for a good selectivity index (SI)? Do you think the VLe with the highest SI has the highest potential to be developed in cancer treatment?
4. Please check the plant name according to the following website: https://wfoplantlist.org/plant-list/taxon/wfo-0001133257-2023-06?page=1
Thank you.
Author Response
Reviewer 1
The manuscript is presented clearly and consists of different analyses that appear to be essential in describing the characteristics of V. boiviniana as well as its potential as antioxidant and in cancer treatment. I have some comments for improvement:
Answer: Thank you for your valuable comments. Your input is highly appreciated.
- Table 1: would you be able to add the information regarding the concentration of each molecule in the each extract? I believe it would be interesting for the readers.
Answer: The qualitative phytochemical profile of the studied species was presented. Unfortunately, the quantitative analysis was not performed, firstly because of the need for authentic standards, and secondly because the QTOF spectrometer is not suitable for the quantification of compounds. To perform quantification, we need to use a triple quadrupole or ion trap spectrometer. However, we will consider quantitative analysis in the future.
- Table 4: In the discussion, please add some information about the interest of choosing the tested enzymes. To which diseases or biological processes are those enzymes linked? What is the interest of inhibiting the activity of such enzymes?
Answer: Some information on the enzyme inhibitory properties was inserted into the revised version.
- Table 5: would it be possible to add statistical analysis to the data? Could you describe in the discussion the criteria for a good selectivity index (SI)? Do you think the VLe with the highest SI has the highest potential to be developed in cancer treatment?
Answer: The statistical analysis was included in Table 5 of the corrected manuscript. The evaluation of statistical significance was conducted in GraphPad Prism using 2-way ANOVA with Dunnett's multiple comparisons test; the CC50 values obtained for cancer cells were compared to the values on non-cancerous VERO cells. Information was also included in the Methodology section. Data available in published literature suggests that SI>3 indicates anticancer selectivity. Thus, VLe meets these criteria. This information was added to the text.
- Please check the plant name according to the following website: https://wfoplantlist.org/plant-list/taxon/wfo-0001133257-2023-06?page=1
Answer: The plant's name was checked with wfoplantlist.org
Thank you.
Reviewer 2 Report
Comments and Suggestions for Authors
Review of manuscript ref. molecules-2685624
Title: Evaluating phytochemical profiles, cytotoxicity, antiviral activity, antioxidant potential and enzyme inhibition of Vepris boiviniana extracts
Authors: Kassim Bakar , Nilofar Nilofar, Andilyat Mohamed, Łukasz Świątek, Benita Hryć, Elwira Sieniawska, Barbara Rajtar, Claudio Ferrante, Luigi Menghini, Gokhan Zengin, Małgorzata Polz-Dacewicz.
General comment: the study reports the phytochemical composition and several bioactive effects of different extracts from Vepris boiviniana. The authors perform an exhaustive chemical characterization of extracts with different solvents from leaves and stem bark of the plant and carry out evaluation of antioxidant capacity by six different methods. Further, they report inhibitory capacity of particular enzymes related to neurodegenerative processes and metabolic unbalance and glycemic control, and they end up with preliminary tests to show the potential cytotoxicity against cancer cells. The hypothesis and objectives are sound and attractive, methods seem adequate and results, although restricted to in vitro and cell culture, are well discussed and with potential translational application after in vivo testing. Some specific comments are detailed below
Specific comments:
1) Lines 290-291 and reference 47; while the text refers to Vepris grandifloria, reference 47 mentions Teclea grandifloria.
2) Table 5 and epigraph 2.5; data depicted in table 5 regarding cytotoxicity and anticancer selectivity of V. boiviniana extracts on cell line FaDu (hypopharyngeal cancer) are not commented nor discussed in epigraph 2.5. In fact, they are not mentioned in the whole text except material and methods.
3) Statistical analysis is missing in the data of table 5.
4) Line 349; it should be mL, capital L.
5) Line 357; it should be Llorent-Martínez et al.
6) Reference 33 is incomplete; journal data is missing.
Author Response
Reviewer 2
Review of manuscript ref. molecules-2685624
Title: Evaluating phytochemical profiles, cytotoxicity, antiviral activity, antioxidant potential and enzyme inhibition of Vepris boiviniana extracts
Authors: Kassim Bakar , Nilofar Nilofar, Andilyat Mohamed, Łukasz Świątek, Benita Hryć, Elwira Sieniawska, Barbara Rajtar, Claudio Ferrante, Luigi Menghini, Gokhan Zengin, Małgorzata Polz-Dacewicz.
General comment: the study reports the phytochemical composition and several bioactive effects of different extracts from Vepris boiviniana. The authors perform an exhaustive chemical characterization of extracts with different solvents from leaves and stem bark of the plant and carry out evaluation of antioxidant capacity by six different methods. Further, they report inhibitory capacity of particular enzymes related to neurodegenerative processes and metabolic unbalance and glycemic control, and they end up with preliminary tests to show the potential cytotoxicity against cancer cells. The hypothesis and objectives are sound and attractive, methods seem adequate and results, although restricted to in vitro and cell culture, are well discussed and with potential translational application after in vivo testing. Some specific comments are detailed below.
Answer: Dear Reviewer, thank you for your kind words and comments, which will enable us to improve our work.
Specific comments:
1) Lines 290-291 and reference 47; while the text refers to Vepris grandifloria, reference 47 mentions Teclea grandifloria.
Answer: Thank you for pointing this out. The accepted species name (according to wfoplantlist.org) is Vepris grandifloria, whereas Teclea grandifloria is a synonym. This information was included in the corrected manuscript.
2) Table 5 and epigraph 2.5; data depicted in table 5 regarding cytotoxicity and anticancer selectivity of V. boiviniana extracts on cell line FaDu (hypopharyngeal cancer) are not commented nor discussed in epigraph 2.5. In fact, they are not mentioned in the whole text except material and methods.
Answer: Thank you for this valuable comment. The information on the toxicity of V. boiviniana leaves infusion (VLi) towards FaDu cells was added in the corrected manuscript.
3) Statistical analysis is missing in the data of table 5.
Answer: The statistical analysis was included in Table 5 of the corrected manuscript. The evaluation of statistical significance was conducted in GraphPad Prism using 2-way ANOVA with Dunnett's multiple comparisons test; the CC50 values obtained for cancer cells were compared to the values on non-cancerous VERO cells. Information was also included in the Methodology section.
4) Line 349; it should be mL, capital L.
Answer: Corrected.
5) Line 357; it should be Llorent-Martínez et al.
Answer: Corrected.
6) Reference 33 is incomplete; journal data is missing.
Answer: Corrected.
Reviewer 3 Report
Comments and Suggestions for Authors The paper represents a significant contribution to the field of botanical research as it delves into an extensive analysis of extracts derived from Vepris boiviniana, shedding light on various critical aspects of these plant-derived substances. The study scrutinizes the phytochemical profiles, cytotoxicity, antiviral activity, antioxidant potential, and enzyme inhibition properties exhibited by these extracts. Employing rigorous and well-established methodologies, the authors have made noteworthy advancements in the phytochemical characterization of Vepris boiviniana in comparison to previously published work. One of the key achievements of this research is the tentative identification of a remarkable spectrum of 60 bioactive compounds. These compounds contribute to the rich diversity of bioactive properties present in Vepris boiviniana. The paper places particular emphasis on these bioactive properties, highlighting their multifaceted nature. Notably, the compounds exhibit antioxidant potential, which is crucial for neutralizing harmful free radicals and mitigating oxidative stress. Additionally, the extracts demonstrate enzyme inhibitory properties, which can have a profound impact on various biochemical processes within the body. The paper's conclusions are firmly grounded in the evidence and arguments meticulously presented throughout the research. The authors have successfully established a logical and well-supported narrative that aligns with the findings of the study. However, the authors also conscientiously acknowledge the need for further investigations. The organization of the paper is good, with figures and tables thoughtfully arranged to enhance the reader's comprehension of the presented evidence. Furthermore, the use of up-to-date and pertinent references bolsters the paper's academic credibility, ensuring that it is firmly anchored within the current body of scientific knowledge.
Author Response
Reviewer 3
The paper represents a significant contribution to the field of botanical research as it delves into an extensive analysis of extracts derived from Vepris boiviniana, shedding light on various critical aspects of these plant-derived substances. The study scrutinizes the phytochemical profiles, cytotoxicity, antiviral activity, antioxidant potential, and enzyme inhibition properties exhibited by these extracts. Employing rigorous and well-established methodologies, the authors have made noteworthy advancements in the phytochemical characterization of Vepris boiviniana in comparison to previously published work. One of the key achievements of this research is the tentative identification of a remarkable spectrum of 60 bioactive compounds. These compounds contribute to the rich diversity of bioactive properties present in Vepris boiviniana. The paper places particular emphasis on these bioactive properties, highlighting their multifaceted nature. Notably, the compounds exhibit antioxidant potential, which is crucial for neutralizing harmful free radicals and mitigating oxidative stress. Additionally, the extracts demonstrate enzyme inhibitory properties, which can have a profound impact on various biochemical processes within the body. The paper's conclusions are firmly grounded in the evidence and arguments meticulously presented throughout the research. The authors have successfully established a logical and well-supported narrative that aligns with the findings of the study. However, the authors also conscientiously acknowledge the need for further investigations. The organization of the paper is good, with figures and tables thoughtfully arranged to enhance the reader's comprehension of the presented evidence. Furthermore, the use of up-to-date and pertinent references bolsters the paper's academic credibility, ensuring that it is firmly anchored within the current body of scientific knowledge.
Answer: Dear Reviewer, thank you for your kind words.